# Bio-Inspired Micromachined Volumetric Flow Sensor with a Big Dynamic Range for Intravenous Systems

**DOI:** 10.3390/s23010234

**Published:** 2022-12-26

**Authors:** Lansheng Zhang, Yingchen Yang, Georgios A. Bertos, Chang Liu, Huan Hu

**Affiliations:** 1ZJU-UIUC Institute, International Campus, Zhejiang University, Haining 314400, China; 2Department of Mechanical Engineering, University of Texas Rio Grande Valley, Edinburg, TX 78539, USA; 3Department of Physical Medicine and Rehabilitation, Feinberg School of Medicine, Northwestern University Prosthetics Orthotics Center, Chicago, IL 60611, USA; 4School of Mechanical Engineering, National Technical University of Athens, 15780 Athens, Greece; 5Applied Sciences and Technology, Baxter Healthcare Inc., Round Lake, IL 60073, USA; 6Institute of Electrics, Chinese Academy of Sciences, Beijing 100089, China; 7State Key Laboratory of Fluid Power and Mechatronic Systems, Zhejiang University, Hangzhou 310027, China

**Keywords:** volumetric flow sensor, haircell, piezoresistor, big dynamic range, intravenous infusion

## Abstract

Real-time monitoring of drug delivery in an intravenous infusion system can prevent injury caused by improper drug doses. As the medicine must be administered into the vein at different rates and doses in different people, an ideal intravenous infusion system requires both a low flow rate and large dynamic range monitoring. In this study, a bio-inspired and micromachined volumetric flow sensor is presented for the biomedical application of an intravenous system. This was realized by integrating two sensing units with different sensitivities on one silicon die to achieve a large dynamic range of the volumetric flow rate. The sensor was coated with a parylene layer for waterproofing and biocompatibility purposes. A new packaging scheme incorporating a silicon die into a flow channel was employed to demonstrate the working prototype. The test results indicate that the sensor can detect a volumetric flow rate as low as 2 mL/h, and its dynamic range is from 2 mL/h to 200 mL/h. The sensor performed better than the other two commercial sensors for low-flow detection. The high sensitivity, low cost, and small size of this flow sensor make it promising for intravenous applications.

## 1. Introduction

Flow rate measurement is crucial in chemical plants, energy, and environmental surveys such as oil monitoring [1,2,3] and ocean-turbulence measurement [4]. Different applications have different requirements regarding the flow speed, sensor size, response time, measurement accuracy, and biocompatibility. Conventional industrial flowmeters usually target large flow rates and are unsuitable for applications that require high-precision measurements of flow parameters, such as flow rate, velocity, and direction [5]. For example, drug delivery [6] and sleep apnea systems [7,8,9] both require high-precision flow-rate control.

In clinical drug therapy, intravenous (IV) injection is a fundamental approach for drug delivery systems. According to medical statistics, up to 80% of patients receive intravenous (IV) therapy during hospitalization. One challenge in the IV system is to precisely monitor a small volumetric flow. An imprecise flow rate or improper drug dose can result in severe injury and death. Currently, most IV systems rely on live monitoring by medical staff, and occasional human errors and miscalculations are inevitable. An infusion pump can improve the accuracy of drug injection; however, its high price limits its applications. In addition, an industrial flowmeter system cannot accommodate the IV system owing to its large size, power consumption, etc.

MEMS technology [10], owing to the advantages of its small size, low cost, and integrating capability, combined with bionics, makes the particular IV flow sensor of this paper possible. MEMS flow sensors fall into two classes. One is the thermal type [11,12], and the other is the non-thermal type. Thermal-type flow sensors are unsuitable for biomedical applications because a heated element can damage the chemical or biological properties of the medicine, causing safety concerns. Non-thermal-type flow sensors include piezoresistive flow [13,14,15], piezoelectric flow [16,17], capacitive flow [18], micro-Coriolis flow sensors [19], etc.

Natural species can provide materials [20], structures [21], and inspiration for scientific research. Biological haircells can be widely found in animals such as fish, insects, amphibians, and humans, which provides inspiration for the development of MEMS flow sensors. Researchers have developed various biomimetic haircell sensors by imitating the hair of crickets [22,23], lateral lines of fishes [24,25,26,27] and amphibian animals [28], etc. We previously demonstrated an artificial haircell (AHC) flow sensor for local-flow velocity measurements [29]. The AHC mimics the haircell in the fish lateral line to detect the water flow, as shown in Figure 1. The AHC sensor consists of an artificial polymer hair placed on the free end of a silicon cantilever. When exposed to a flow, the artificial hair experiences a drag force that bends the silicon cantilever beam and generates stress, as schematically illustrated in Figure 2. Thereafter, a stress-sensitive resistor made by boron implantation changes its resistance upon stress to accurately sense the local flow velocity with the assistance of amplifying circuits.

Here, the design, fabrication, and characterization of a small, low-cost, and high-sensitivity volumetric flow sensor based on an AHC sensor are presented. The sensor aims to provide feedback on the drug-delivery rate in the IV system, and holds the potential for measuring small volumetric flow rates in other applications. This MEMS volumetric flow sensor could realize a miniaturization system and provide sensing means in microfluidics and other microenvironments where it is difficult to accommodate macro-size flow sensors.

## 2. Sensor Design

### 2.1. Sensor Design

Figure 3 shows the proposed design scheme for the volumetric flow sensor. It consisted of a custom-built flow module with a silicon die that incorporated two AHCs. These two AHCs were designed with different sizes to achieve flow sensing over a wide dynamic range. A through-wafer design was employed in the AHC fabrication to enable an electrical connection from the top of the silicon base to the bottom to ensure a smooth top surface for minimal flow disturbance. The flow channel had a circular cross-section at the inlet and outlet, but gradually and smoothly adapted the shape to a square cross-section in the middle portion where the sensing die was placed. The entire structure was coated with parylene film to make it waterproof and drug-compatible [30,31].

### 2.2. DC Flow Hydrodynamic Mode

A fully developed two-dimensional laminar flow through a square cross-sectional conduit was assumed in this model. The flow direction was considered perpendicular to the longitudinal axis of the hair by neglecting the hair deflection angle under hydrodynamic loading. The hair had a cylindrical shape with a uniform cross-sectional area and finite length. The moment M exerted on the hair in a steady-state laminar flow was estimated using a local drag coefficient approach, as discussed below.

#### 2.2.1. Local Flow Profile and Volumetric Flow Rate

For a fully developed laminar flow in a rectangular conduit with dimensions of −a≤y≤a,
−b≤z≤b in the (y,z) plane, the flow velocity u along the conduit (*x*-axis) as a function of y and z can be determined using [32]:(1)$$u\left(y,z\right)=\frac{16a^{2}}{\pi^{3}\mu }\left(-\frac{d\hat{p}}{dx}\right)\overset{\infty }{\underset{i=1,3,5,\ldots }{\sum }}{\left(-1\right)}^{\frac{i-1}{2}}\left[1-\frac{\cos h\left(\frac{i\pi z}{2a}\right)}{\cos h\left(\frac{i\pi b}{2a}\right)}\right]\frac{\cos\left(\frac{i\pi y}{2a}\right)}{i^{3}}\;,$$
where μ is the fluid viscosity and $-d\hat{p}/dx$ is the pressure gradient along the conduit. The pressure gradient can be resolved using a specified maximum velocity at the centerline of the conduit. Then, the volume flow rate $Q_{0}$ can be evaluated using [32]
(2)$$Q_{0}=\frac{4ba^{3}}{3\mu }\left(-\frac{d\hat{p}}{dx}\right)\left[1-\frac{192a}{\pi^{5}b}\overset{\infty }{\underset{i=1,3,5,\ldots }{\sum }}\frac{\tan h\left(\frac{i\pi b}{2a}\right)}{i^{5}}\right]\;.\;$$

For the conduit of a square cross-section, as shown in Figure 3b, a=b=1 mm.

#### 2.2.2. Strain Estimate of the Cantilever Beam

The hair was regarded as a uniform cylinder in the flow, as shown in Figure 4. The cylindrical hair of diameter d can be divided into N small segments with length Δh. For a small segment i (i=1, 2, …, N) in a flow with density ρ, velocity $u_{i}$, and drag coefficient $C_{D\_i}$, the segmental drag force $F_{D\_i}$ can be evaluated using [29]
(3)$$F_{D\_i}\approx \frac{1}{2}C_{D\_i}\rho u_{i}{}^{2}d\Delta h.$$

The drag coefficient $C_{D\_i}$ is related to the local Reynolds number $Re_{i}$ within each segment. For $Re_{i}$ < 10, $C_{D\_i}$ can be estimated using
(4)$$lnC_{D\_i}\approx -0.67lnRe_{i}+2.51.$$

The local Reynolds number is defined as
(5)$$Re_{i}=\frac{\rho u_{i}d}{\mu }.$$

As shown in Figure 4, the flow velocity $u_{i}$ varied from segment to segment. This could be determined using Equation (1). Subsequently, with the segmental drag determined using Equation (3), the total torque about the hair root could be calculated through integration as follows:(6)$$M=\overset{N}{\underset{i=1}{\sum }}F_{D\_i}\Delta hi.$$

With the total torque obtained, the resulting strain on the cantilever beam could be determined using
(7)$$\varepsilon =\frac{Mt}{2EI}\;,$$
where t is the beam thickness, I is the area moment of inertia, and E is Young’s modulus of the silicon cantilever.

The above equations provide a rough estimation of cantilever strain. The analysis is based on an ideal laminar flow, which may not precisely reflect the actual flow conditions. The hair also deflects in the flow, lowering the cilia height. Different packaging methods can alter the local flow conditions to a certain extent. Regardless of such deviations from the assumed conditions, the above-discussed analytical approach can still serve as an effective tool for examining the parametric effect and guiding the sensor’s design.

## 3. Fabrication

To integrate the flow sensor into a flow chamber to enable volumetric flow sensing, a through-silicon via (TSV) design was employed in this study. It compares to the previous packaging scheme of the wire-bonding of the electrical pads on the front surface of the silicon die to a printed circuit board (PCB) board where the silicon die sits [26]. In this earlier packaging, the bonding wires were encapsulated with epoxy to prevent contact with liquid. Consequently, the epoxy formed a large hump on the front surface, which in turn altered the local flow condition to a certain extent owing to flow interaction. The TSV design was intended to create a much smoother front surface for a less-disturbed local flow. Four TSVs were integrated into the fabrication process to enable an electrical connection from the backside of the sensor dies to the front surface. The entire fabrication process of the volumetric sensor consisted of two essential steps: sensor fabrication and packaging.

### 3.1. Sensor Fabrication

The MEMS micromachining process used an SOI wafer with 2 μm-thick top silicon, 2 μm-thick buried oxide, and 400 μm-thick bottom silicon layers. Figure 5 illustrates the major fabrication steps. The fabrication began with selective boron doping on an oxidized and patterned SOI wafer to produce piezoresistors (Figure 5a). A boron beam of 60 KeV energy was applied to achieve a target concentration of 1 × 10^20^ cm^−3^. Further, an oxide layer was grown and selectively opened to make contact windows through lithography (Figure 5b). Subsequently, aluminum electrodes were fabricated through lithography, aluminum evaporation, and lift-off to make electrical contact with the piezoresistors (Figure 5c). With an additional lithography step followed by topside deep reactive ion etching (DRIE) using the Bosch process [33], the cantilever and TSV shapes were defined (Figure 5d). Subsequently, the lithography on the bottom of the wafer and DRIE etching were performed to etch the cantilever-release cavity and TSVs (Figure 5e). In the next step, SU-8 hair was formed on the topside through the lithography of a thick SU-8 film (Figure 5f). Finally, the cantilever was released by etching the silicon oxide away in a buffered oxide etchant (BOE, 1:10 dilution) (Figure 5g). Figure 6 shows the completed volumetric flow sensor.

### 3.2. Sensor Packaging

As shown in Figure 5h, the PCB board vias were matched with through-wafer holes on the silicon die. Each via was threaded with an electrical wire, and silver paste was applied between the wires and front electrode pads to establish electrical connections. Figure 7 shows a photograph of the sensor die after silver paste was applied.

After the wire connection was formed between the sensor die and the PCB board, the formed unit was coated with parylene for waterproofing. Parylene is a biocompatible material that minimizes its influence on medicine [34]. The unit was then embedded in the flow chamber, as shown in Figure 5i. The flow chamber was fabricated by rapid prototyping. Figure 8 shows two photos of the completed device with and without a water flow.

## 4. Characterization and Discussion

Figure 9 shows the setup used to characterize the volumetric flow sensor. A syringe pump (Havard 1000) was employed to generate a water flow with a specific volumetric flow rate, which was connected through a pipe to the inlet of the flow chamber. The outlet of the flow chamber was connected to a beaker filled with water to provide constant back pressure. The sensor was electrically connected to a signal-processing circuit, as illustrated in Figure 10. The circuit consisted of four parts. The first part was a voltage regulator that provided a constant 1.5 V voltage supply with reduced voltage fluctuation. The second part was a resistor network that balanced the output of a Wheatstone bridge by adjusting a zero-pot. The third part was the Wheatstone bridge, which consisted of one piezoresistor on the silicon die cantilever and three other piezoresistors not on the silicon cantilever, but close to it, for noise suppression. The fourth part was an amplifier INA 126 with a gain of 405.

It was found that the supply voltage should be controlled at a relatively low level, because a high voltage would noticeably heat the four piezoresistors unevenly. The piezoresistor on the cantilever had a different thermal-conducting condition than the other three. Therefore, heating would result in different temperatures among the four piezoresistors, creating thermal noise as well as drift.

Different volumetric flow rates were generated by the syringe pump in the range of 2–400 mL/h. Appendix A shows the sensor output acquired by an oscilloscope at volumetric flow rates of 100 mL/h (Appendix A) and 200 mL/h (Appendix A). The syringe pump was turned on and off twice at each flow rate to generate repeated signal patterns. As shown in Appendix A, the output remained nearly zero at the beginning because the syringe pump was not started. Subsequently, the output gradually increased as the flow rate increased from zero to a preset value, followed by a nearly constant value at a relatively stable flow rate. The syringe pump was maintained at this volumetric flow rate for seconds and then turned off. In response, the sensor output decreased to almost zero. Appendix A shows a similar pattern to that shown in Appendix A, but at a higher peak level owing to a high flow rate.

The systematic test results of the volumetric flow sensor under various flow rates are presented in Figure 11 and compared with the simulation results. The sensor output began to saturate when the volumetric flow rate was close to 200 mL/h, and the corresponding Reynolds number was approximately eight. The reason for this saturation is not fully understood. One hypothesis is that the pressure drop across the haircell sensor increases with an increasing flow rate, which leads to a gradually strengthened branch flow passing through the cavity underneath the cantilever beam. Such a branch flow creates stress on the cantilever beam that is opposite to that generated by the haircell and thus cancels a portion of the signal output. Such a canceling effect results in signal saturation to a certain threshold of the flow rate.

A common method in IV systems is to employ a conventional weight sensor to monitor the remaining drug volume. However, volume calculations would be inaccurate for small-volume doses. Another commonly used method is to monitor the drip rate using an infrared sensor [35]. However, optical sensors are relatively expensive and vulnerable to errors owing to the external disturbances of light and temperature. Other methods have also been reported for the IV systems. A flexible capacitive sensor integrated with a wireless communication module was used to monitor the status of the fluid bag [36]. An IV system with a volumetric flow rate (0.05 mL/min) using a capacitive MEMS ultrasound sensor was reported [37]. Currently, Internet-of-Things (IoT)-enabled drip-monitoring systems have been reported in the literature [38,39].

An accurate and suitable flow rate in IV systems is indispensable because the medicine must be administered into the vein at a safe rate in different people [40]. According to previous reports, with no or limited oral intake, a full-term neonate who receives a total fluid allowance of 100–140 mL/kg/day will typically receive intravenous fluids at rates of 10–20 mL/h. Furthermore, for extremely low birth weight (<1000 g) neonates, infusion rates are often between 3 and 5 mL/h [41]. In this case, some commercial flow sensors, such as Shift Labs’ Drip Assist and Sensirion LD20 (threshold detection limit ~5 mL/h), are not applicable [42]. In other conditions, the flow rate in adult IV systems is 45–60 drops per minute [43], approximately 135–180 mL/h (20 drops per ml). This indicates that the flow rate of the IV system varies over a large range between adults and neonates. However, various flow sensors in IV systems cannot handle both a low threshold limit and large dynamic range detection [44,45]. Both a low threshold detection limit and a large dynamic range are critical for constructing a widely applicable IV system. The haircell flow sensor has a lower threshold detection limit (2 mL/h) and a large dynamic range from 2 mL/h to 200 mL/h, which performs better in IV system applications. Moreover, owing to MEMS microfabrication technology, this haircell flow sensor could be much cheaper in the current commercially available flow sensor market.

## 5. Conclusions

A bio-inspired volumetric flow sensor that mimicked a hair cell sensor with a sensitivity of 2 mL/h was developed. It also exhibited a detection range of 2 mL/h to 200 mL/h. A new packaging scheme was employed by incorporating through-wafer vias into a sensor die to achieve a hydrodynamically preferred smooth front surface. The sensor was compared with two commercial IV sensors (5 mL/h), which showed that the sensor performed lower threshold detection (2 mL/h). It could also achieve monolithic integration with a CMOS integrated circuit to provide small, low-cost, and low-power sensing. The low-flow-rate-detection ability and biocompatibility advantages indicate that the sensor has promising applications in genomics, point-of-care diagnostics, and drug-delivery systems, such as IV systems, for monitoring tiny volumes of medicines. Further research will improve sensor packaging to achieve a more stable signal and apply the sensor to more microfluidic systems.

## Figures and Tables

**Figure 1 sensors-23-00234-f001:**
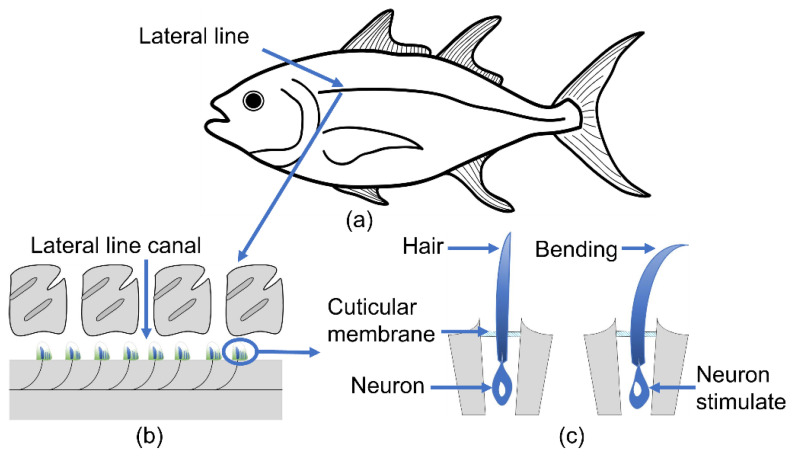
Schematic of bionic principle. (**a**) Fish lateral line; (**b**) basic unit of the lateral sensory organ; (**c**) haircell sensing principle.

**Figure 2 sensors-23-00234-f002:**
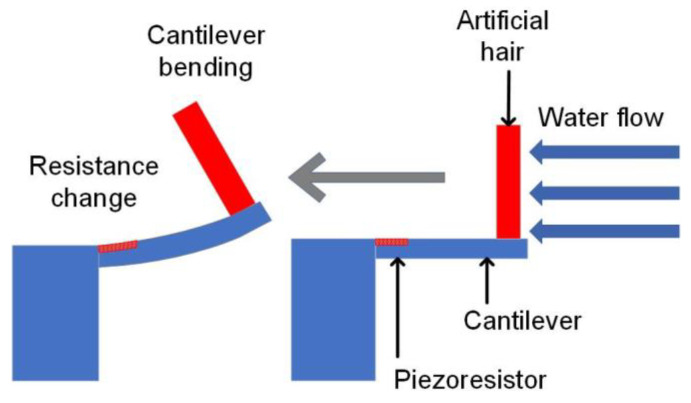
A schematic drawing illustrates the working principle of the AHC flow sensor.

**Figure 3 sensors-23-00234-f003:**
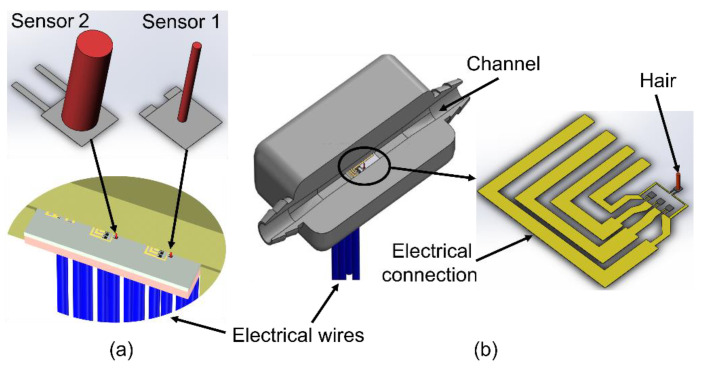
Schematics of the volumetric flow sensor. (**a**) Two AHCs on one silicon die; (**b**) cutaway view of the sensor assembly and a close-up of an individual AHC.

**Figure 4 sensors-23-00234-f004:**
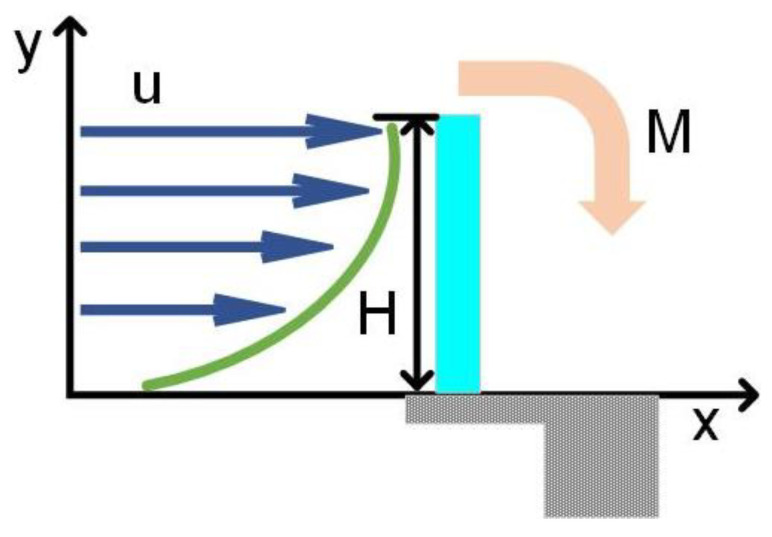
Schematic describing that a hair structure in a steady flow experiences a momentum generated by the drag force exerted on the hair.

**Figure 5 sensors-23-00234-f005:**
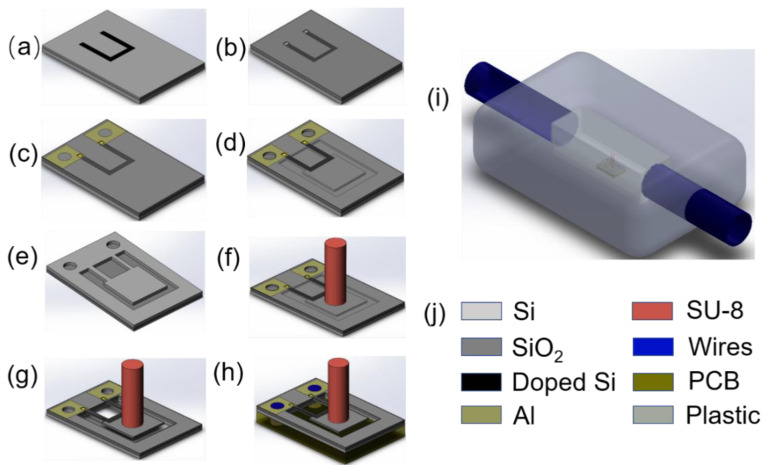
Fabrication process of the volumetric flow sensor. (**a**) Piezoresistive formation by boron ion injection; (**b**) Making contact windows; (**c**) Evaporation to make aluminum electrode; (**d**) Top silicon dioxide etching and then silicon etching; (**e**) Backside etching; (**f**) SU-8 hair process; (**g**) Cantilever release; (**h**) Bonding to a PCB board; (**i**) Embedding into the flow channel; (**j**) Definitions of colors used in (**a**) through (**i**).

**Figure 6 sensors-23-00234-f006:**
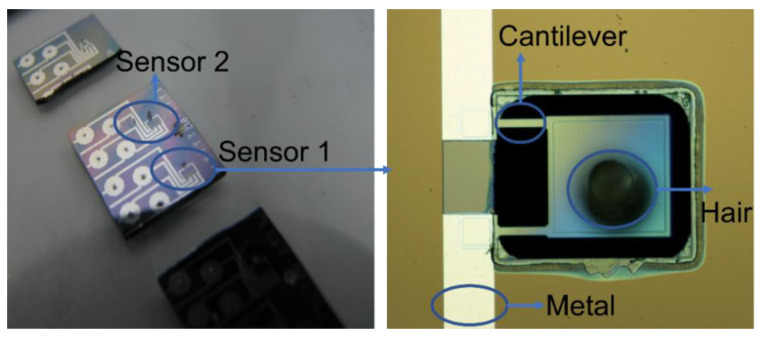
Volumetric flow sensor and a scanning electron micrograph of sensor 1.

**Figure 7 sensors-23-00234-f007:**
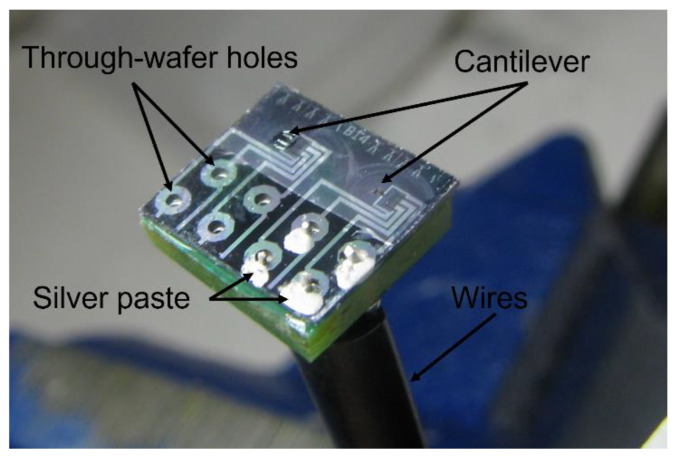
Photo showing two sensors glued on a PCB board.

**Figure 8 sensors-23-00234-f008:**
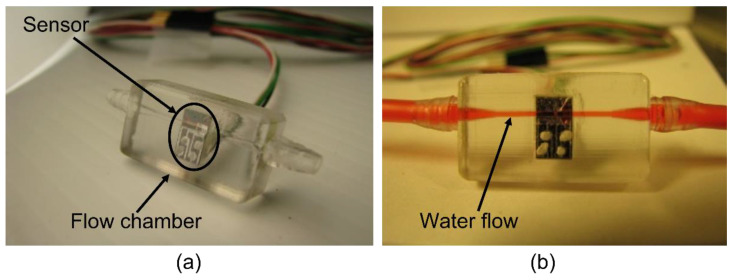
(**a**) Assembled volumetric flow sensor with sensing hairs in the middle of the fluid passage. (**b**) Colored water runs through the fluid passage of the sensor.

**Figure 9 sensors-23-00234-f009:**
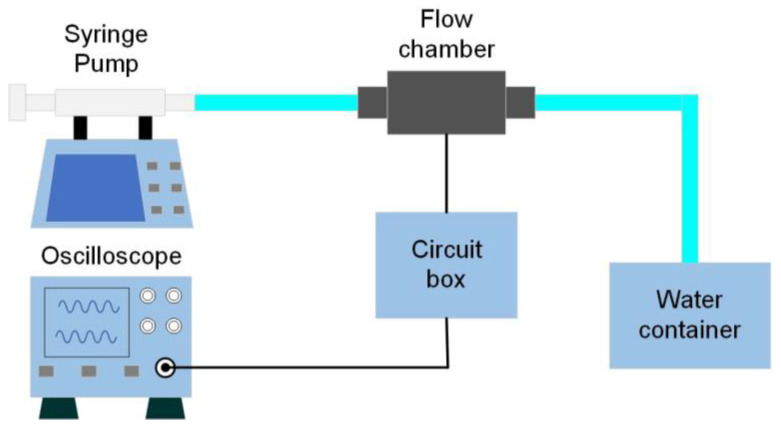
Schematic of the experimental setup for characterization of the volumetric flow sensor.

**Figure 10 sensors-23-00234-f010:**
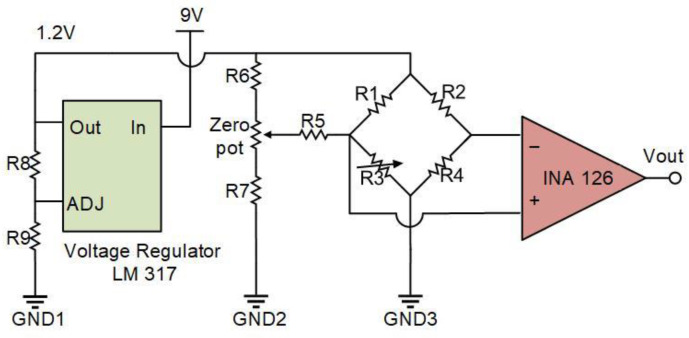
Diagram of the circuit box that interfaces with the sensor and amplifies the output signal.

**Figure 11 sensors-23-00234-f011:**
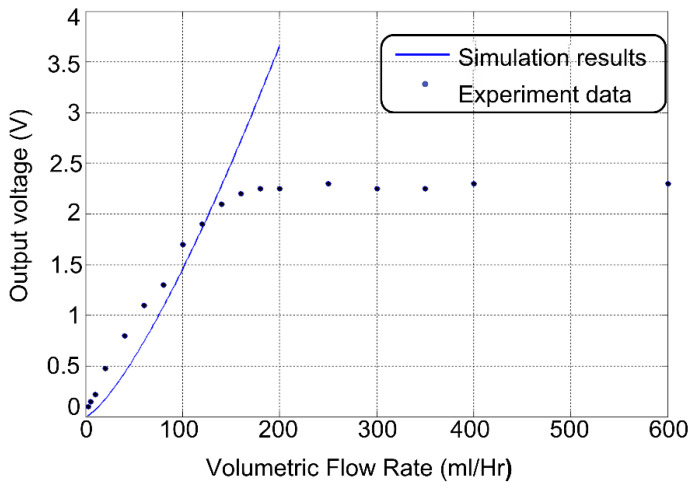
Output voltage versus the volumetric flow rate ranges between 2 mL/h and 400 mL/h.

## Data Availability

The raw data supporting the conclusion of this article will be made available by the authors, without undue reservation.

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
