# Peer review of "Bio-Inspired Micromachined Volumetric Flow Sensor with a Big Dynamic Range for Intravenous Systems"

_sensors, 2022, doi:10.3390/s23010234_

Round 1

Reviewer 1 Report

This is a paper describing a mems volumetric flow sensor, giving design, fabrication, and testing information. I find it well prepared and a valuable contribution. Here are some minor points to consider:

1. It is not clear what are the black posts under the silicon die in Figure 3?

2. Above Eq. 1, why the two 2's in 2aX2b? And given that they are needed should not there be a 2 on the b in the cosh term of (1)?

3. A reference to Bosch process on line 160 of page 5 would help.

4. Line 213 mentions Figure S1 which is not present but probably should be.

5. What are typical values for the resistors in Figure 10? 

Reviewer 2 Report

Reviewer Reports:

I recommend a major amendment at this level.

General comments:

The manuscript entitled A bio-inspired micromachined volumetric flow sensor with a big dynamic range for intravenous systems” was reviewed. The work carried out in the manuscript is interesting and aimed at demonstrating a small-size, low-cost, and high-sensitivity volumetric flow  sensor based on this AHC sensor that can be used in IV systems to provide feedback on the drug delivery rate. The manuscript has a lot of information however, there are some lacking connectors and the writing style makes it very confusing. It is better to do not to use the first-person's pronoun. Do not use "we, us, or our" throughout the paper. The authors are suggested to proofread the paper with a native English speaker and restructuring of sentences is required for the entire manuscript. Better connect your research findings to previous works published in sensors and in other top journals. Would you explicitly specify the novelty of your work? What progress against the most recent state-of-the-art similar studies was made? Additionally, the novelty of the research still is not clear and the discussion and conclusions can not satisfy me. The innovation and the importance of this work are not clearly highlighted in the abstract, introduction and conclusions. Please work on this and prove to us why this work is valuable. The authors do not validate the obtained results and compare them with the other works in the field. The authors need to present how the results can be validated and verified. Please also remove ANY lumped references. Please define each of them separately to avoid inappropriate citations. Too many abbreviations are used in the analysis and results. I recommend a nomenclature section for the abbreviations and variables used throughout the passage. Other main remarks that in my opinion needs attention are the following:

Detailed comments:

Title:

Ok.

Abstract:

Too brief. The abstract does not work well. A good abstract should address these issues: what are you trying to do, why, what you found and what is the significance of your findings? In the abstract, please add an indication of the achievements from your study that are relevant to the journal's scope. The abstract should state briefly the purpose of the research, the principal results and major conclusions. An abstract is often presented separately from the article, so it must be able to stand alone. In the abstract, please add an indication of the achievements from your study that are relevant to the journal's scope. Please be concise - maximum 1-2 lines.

Introduction:

The literature review should clarify the "contribution" of your study. The authors failed to present the study debates and failed to discuss the debates. In general, the authors should present the specific debate for your study. This should more clearly show the knowledge gaps identified and link them to the paper's goals. A high-quality paper has to provide a proper state-of-the-art analysis after the literature review and only based on the analysis to formulate the paper's goals. The lack of proper justification creates the wrong impression that the authors are unaware of the recent developments. The relevant reference may be of interest to the author according to below:

http://www.jett.dormaj.com/docs/Volume8/Issue%203/Synthesis%20of%20Silver%20Nanoparticles%20from%20Fish%20Scale%20Extract%20of%20Cyprinus%20carpio%20and%20its%20Decolorization%20Activity%20of%20Textile%20Dyes.pdf

Please eliminate the use of redundant words. Eg. In this way, Recently, Respectively, therefore, currently, thus, hence, finally, to do this, first, in order, however, moreover, nowadays, today, consequently, in addition, additionally, furthermore. Please revise all similar cases, as removing these term(s) would not significantly affect the meaning of the sentence. This will keep the manuscript as CONCISE as possible. Please check ALL. Avoid beginning or ending a sentence with one or a few words, they are usually redundant. Kindly revise all.

Results and Discussion:

All the findings of the current work need to be compared and discussed with the results of other researchers finding instead of having a general comparison with other researchers' works. The authors should perform a comparison between the forecasting results. In your discussion section, please link your empirical results with a broader and deeper literature review.

Conclusions:

In the conclusions, in addition to summarising the actions taken and results, please strengthen the explanation of their significance. It is recommended to use quantitative reasoning compared with appropriate benchmarks, especially those stemming from previous work.

References:

Please check the reference section carefully and correct the inconsistency.

Round 2

Reviewer 2 Report

Reviewer Reports:

I have reviewed the revised version manuscript entitled" A bio-inspired micromachined volumetric flow sensor with a big dynamic range for intravenous systems”. The paper has been improved and can be accepted.